# TaMMa: Target-driven Multi-subscene Mobile Manipulation

**Jiawei Hou**[*], **Tianyu Wang**[*], **Tongying Pan**[*], **Shouyan Wang,**
**Xiangyang Xue, Yanwei Fu**[†]

**Abstract:** For everyday service robotics, the ability to navigate back and forth based on tasks in multi-subscene environments and perform delicate manipulations is crucial and highly practical. While existing robotics primarily focus on complex tasks within a single scene or simple tasks across scalable scenes individually, robots consisting of a mobile base with a robotic arm face the challenge of efficiently representing multiple subscenes, coordinating the collaboration between the mobile base and the robotic arm, and managing delicate tasks in scalable environments. To address this issue, we propose Target-driven Multi-subscene Mobile Manipulation (*TaMMa*), which efficiently handles mobile base movement and fine-grained manipulation across subscenes. Specifically, we obtain a reliable 3D Gaussian initialization of the whole scene using a sparse 3D point cloud with encoded semantics. Through querying the coarse Gaussians, we acquire the approximate pose of the target, navigate the mobile base to approach it, and reduce the scope of precise target pose estimation to the corresponding subscene. Optimizing while moving, we employ diffusion-based depth completion to optimize fine-grained Gaussians and estimate the target's refined pose. For target-driven manipulation, we adopt Gaussians inpainting to obtain precise poses for the origin and destination of the operation in an *imagine before you do it* manner, enabling fine-grained manipulation. We conduct various experiments on a real robotic to demonstrate our method in effectively and efficiently achieving precise operation tasks across multiple tabletop subscenes.

**Keywords:** Multi-subscene, 3D Gaussians, Scene Inpainting, Target-driven Mobile Manipulation

## 1 Introduction

It is common and highly practical for robotics to perform everyday tasks with cross-subscene mobility and fine-grained manipulation in a scalable environment. Handling compositional tasks is especially challenging. For example, robotics may be required to pick up a cup from one table and place it on another, pour water from a bottle in one location into a cup in another, collect cups from various tabletops and organize them on a cup rack, or tidy up multiple surfaces. However, existing robotics mainly focus on complex tasks in a single scene [1, 2, 3, 4] or simple tasks in a scalable scene [5, 6, 7, 8, 9] individually. Previous successful works [10, 8, 9, 11, 12, 13] often relied on Large Multimodal Models (LMMs) for scene perception and action planning, laying the groundwork for accomplishing long-horizon tasks. Nevertheless, due to a lack of further refinement of 3D scenes and their action planning often being in the form of policies, their generalization ability to scenes is relatively weak, leading to low success rates in various compositional tasks. Works like [5, 14] have also employed iterative search through building 3D scene graphs to accomplish

---

[*] Equal contribution.

[†] Corresponding author.

Jiawei Hou, Tongying Pan and Xiangyang Xue are with School of Computer Science, Fudan University; Tianyu Wang and Shouyan Wang are with Institute of Science and Technology for Brain-Inspired Intelligence, Fudan University; Yanwei fu is with School of Data Science, Fudan University. {*jwhou23, tywang22 typan23*}*@m.fudan.edu.cn* {*xyxue, shouyan, yanweifu*}*@fudan.edu.cn*

Dr. Fu and Dr. Xue are also with Fudan ISTBI—ZJNU Algorithm Centre for Brain-inspired Intelligence, Shanghai Key Lab of Intelligent Information Processing, and Technology Innovation Center of Calligraphy and Painting Digital Generation, Ministry of Culture and Tourism, China.

This work was supported in part by Shanghai Platform for Neuromorphic and AI Chip under Grant 17DZ2260900 (NeuHelium).

8th Conference on Robot Learning (CoRL 2024), Munich, Germany.

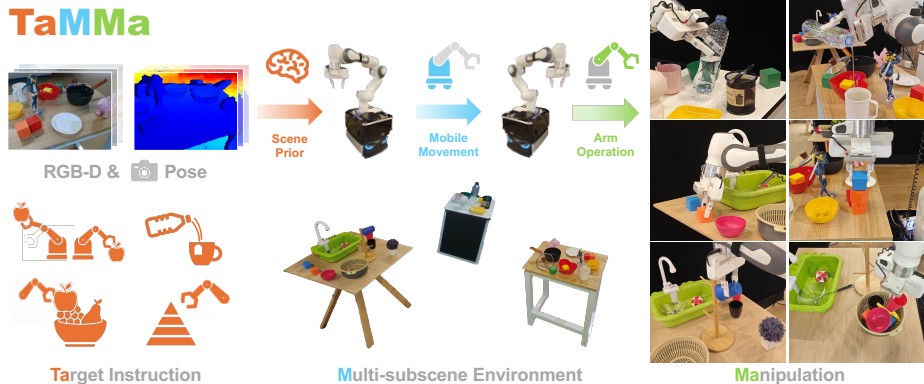

Figure 1: Our proposed multi-subscene mobile manipulation approach. Coarse scene prior and refined pose estimation empower the mobile base movement and robotic arm manipulation effectively and efficiently on compositional challenging tasks such as pouring, stacking, and tidy-up.

tasks across multiple subscenes. However, the computational resources and manpower required for this approach are substantial. It is essential to research a method that can construct unstructured 3D subscenes and execute precise manipulations to handle compositional tasks efficiently.

For the sake of generality and simplicity, we assume the robotics structure is conducted with a robotic arm leveraging on a mobile base. To tackle cross-subscene compositional target-driven mobile manipulation tasks, robotics need the following properties: (1) Represent precise multiple subscenes in a unified space. In such multi-subscenes, it is impractical to achieve cross-subscene tasks with observation through a single camera, ignoring the environmental prior knowledge. (2) Collaborate the mobile base with the robotic arm. The end-to-end robotic learning pipeline is not ready to solve delicate cross-subscene tasks due to the challenge of generalizing to different scenes and hardware. (3) Obtain precise pose for target manipulation. To achieve compositional target-driven mobile manipulation in a scalable scene, estimating a fine-grained 6 Degree-of-Freedom(DoF) pose is equally significant besides representing the whole scene.

To effectively represent 3D scenes and obtain the pose of target objects, the distilled feature fields(DFFs) like works[15, 16, 17, 18, 19, 20, 21] have recently been proposed to model 3D scenes with implicit representations by distilling 2D features to 3D, enabling scene content querying and interaction. However, these approaches suffer from (1) the requirement of sufficient views to train the feature field, (2) the disability to perform complex tasks such as pouring liquids or stacking, (3) the lack of generalizability for tasks across multiple sub-scenes, and (4) the disability to model environmental changes caused by manipulation. At the same time, 3D Gaussian Splatting(3DGS)[22] is proposed as a highlight approach with real-time processing capability that enables scene editing. The application of diffusion-based strategies has achieved promising results [23, 24, 25], demonstrating its few-shot reconstruction and scene completion ability. Recently, InFusion[26] proposed using diffusion priors for 3D Gaussians editing. While the 6 DoF pose of targets serves as a powerful guide for precise manipulation, these approaches enable the possibility of representing and editing scenes efficiently in a scalable scene to get a refined target pose at the origin and destination.

In this paper, we propose *TaMMa* to realize cross-subscene fine-grained target-driven mobile manipulation. Our approach leverages scene priors to facilitate mobile movement and simplify subsequent pose optimization. We employ a diffusion-based 3D Gaussian completion and inpainting technique to obtain the refined target pose for manipulation. Specifically, we first scan only a few posed RGB-D frames of each subscene and unproject them to a sparse 3D point cloud to initialize the 3D Gaussians. Grounded-Light-HQ-SAM[3] is introduced to extract queries and encode semantics following [27, 28] into Gaussians, which can be queried to provide a coarse pose of the manipulation target. This coarse pose serves as a prior, deciding which subscene the target belongs to. The robotic base is then navigated to approach the target, and the scope of fine-grained pose estimation is narrowed down to the related subscene. Simultaneously, the diffusion-based depth completion and

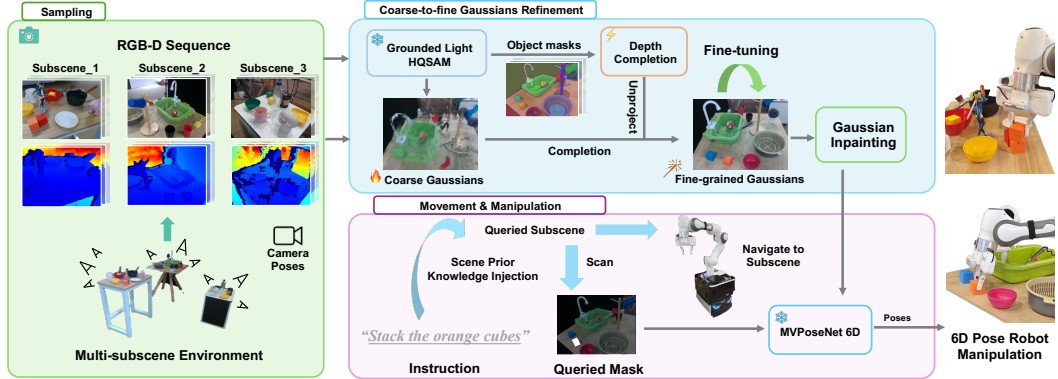

Figure 2: The pipeline of our proposed cross-subscene mobile manipulation method. RGB-D sequences with camera poses are used for point cloud unprojection, which initializes the Gaussians. Grounded-Light-HQ-SAM[3] is employed to encode semantics for Gaussians and depth completion. The coarse Gaussians provide a scene prior to mobile base navigation; further, a fine-grained manipulation target pose is optimized from completed and inpainted Gaussians with an object mask.

Gaussian refinement are processed, and a precise pose is optimized from the refined Gaussians and target inpainting for fine-grained manipulation. The method pipeline is shown in Fig. 2.

In summary, our contributions are as follows: (1) To facilitate robotics in performing everyday tasks, we propose *TaMMa*, showing showcases cross-subscene compositional target-driven mobile manipulation ability efficiently and effectively. (2) By fusing scene-level priors from roughly reconstructed 3D Gaussians and precisely optimized target pose from diffusion-powered Gaussians completion and inpainting, *TaMMa* realizes cross-subscene movement and manipulation with insufficient observations in a sophisticated environment. (3) We conduct various experiments on Franka Panda arm with a mobile base, evaluating the availability of the 3D Gaussians reconstruction and refinement, the scene inpainting, and the fine-grained cross-subscene manipulation.

## 2 Related Works

**3D Scene Reconstruction**. To obtain accurate object poses for robotic manipulation, it is necessary to model the 3D scene with sufficient geometric and semantic information. Neural radiance field(NeRF)[29], as a highly prominent work in scene reconstruction, has attracted significant attention since its proposal, encouraging impressive works[30, 31, 32, 33, 34, 35]. Recently, DFF approaches[15, 16, 17] distilled features into a 3D implicit feature field, achieving a unified 3D feature representation. However, they fall short in the requirement of sufficient views, the disability to perform complex tasks, the lack of generalizability for multiple sub-scenes, and the disability to model environmental changes. On the other hand, 3D Gaussian Splatting[22] has gained widespread attention due to its promising reconstruction results and fast training process. However, the reconstruction performance of 3D Gaussians heavily relies on the initial points obtained from Structure from Motion(SFM)[36]. Inadequate initialization can lead to overfitting on training observations. InFusion[26] employed depth completion learned from diffusion models, demonstrating that with a learned depth inpainting model, the placement of initial Gaussian points can be refined obviously. These works showcase the possibility of completing and inpainting the scene representation with inadequate observations of a scalable scene.

**Depth-only Pose Estimation Preliminaries**. Obtaining the 6-DoF pose of objects serves as a powerful guide for executing precise robotic manipulation. We have made some extensions to the existing pose estimation methods [37, 3, 38] based on grounded visual perception, which can handle common tabletop-level category objects. The extended method relies solely on depth map inputs and aligns shapes using the queried mask with template classes, enabling pose estimation adaptable to unstructured settings. In real-world robotic perception, the quality of point clouds obtained from typ-

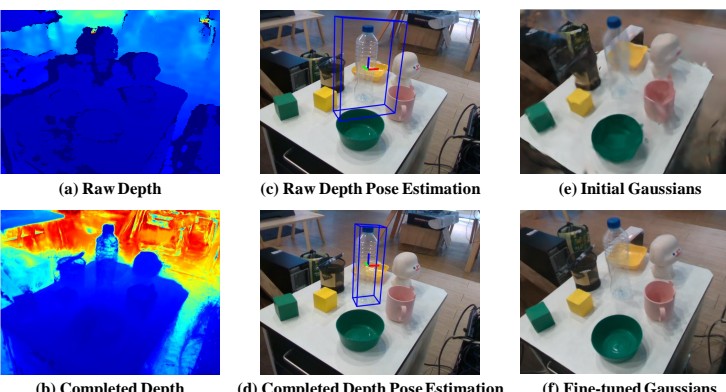

| (a) Raw Depth | (c) Raw Depth Pose Estimation | (e) Initial Gaussians |
| (b) Completed Depth | (d) Completed Depth Pose Estimation | (f) Fine-tuned Gaussians |

Figure 3: The comparison of depth completion effects. (a) is the raw depth shot by RealSense camera. (b) is the diffusion-based completed depth. Pose estimation in (c) is conducted on the raw depth, and in (d), it is leveraged on the depth from completed Gaussians. (e) shows the initial Gaussians and (f) shows the Gaussians fine-tuned by diffusion-completed depth.

ical depth cameras is often unsatisfactory, making depth estimation challenging for semi-transparent and highly reflective metallic objects. This significantly limits depth-based pose estimation models. In this work, we further harness the ability of latent diffusion models (LDMs) [24, 39, 25] to complete depth and enhance real-world pose estimation by optimizing the depth map.

**Target-driven Mobile Manipulation**. Target-driven mobile manipulation (TDMM) has emerged as a critical area in robotics, aiming to enhance the capabilities of robotics in complex environments. The primary goal of TDMM is to enable mobile robotics to navigate and interact with objects to achieve specific tasks or targets, often in unstructured settings. This requires a synthesis of advanced techniques in visual perception, navigation, and manipulation [40]. Such synthesis presents numerous challenges. First, target-driven manipulation requires a fine-grained perception of the interactive scene and continuous guidance of robotic manipulation through foreseeable goals. CLIP-Fields [41] proposed a neural field which can be queried for mobile movement. Previous work [5, 14, 6, 42] has integrated global and local perception information through 3D scene mapping, providing executable plans for mobile manipulator robotics. Although some existing real-world scene maps from these works can understand and represent the environment, they are often structured. Xiong et al. [43] proposed an adaptive learning framework to tackle realistic articulated object operation. These works mainly present two issues: 1. *The mapping process of the task space is time-consuming;* 2. *It fails to meet the needs for fine-grained scene updates.* Second, mobile manipulation tasks often require the integration of navigation and actions to provide target-oriented sequences. Benefiting from the emergent capabilities of large language models (LLMs) in long-horizon planning [44, 45, 46], some works have decomposed mobile manipulation into a series of atomic tasks. However, they have not adequately considered the affordance of real-world objects, resulting in low accuracy in achieving the goals. On the other hand, imitation learning from human demonstrations has shown impressive performance in robotics. Works like MOMA-Force[47], TeleMoMa[48], Mobile-ALOHA[49], and [50] employed behavior cloning on solving mobile manipulation tasks. However, data collection and demonstration selection which rely on complex human labor remain a problem. In this work, we address these challenges by integrating 3D Gaussians and category-level depth-only pose estimation algorithms and focusing on the environment and object representations. Additionally, we edit scenes via 3D Gaussian inpainting, providing operational targets that can be utilized across different scenes.

## 3 Method

### 3.1 Formulation and Overview

**Problem Formulation** Given an unstructured environment with multiple subscenes $\mathcal{S} = \{s_1, s_2, \ldots, s_m\}$, we aim to achieve a cross-subscene instruction $T$ of the target task leveraging on

a mobile base and robotic arm with a set of color images $\mathcal{I} = \{I_i\}_{i=1}^n$, depth images $\mathcal{D} = \{D_i\}_{i=1}^n$ and relative poses $\mathcal{P} = \{P_i\}_{i=1}^n$ as input. To be specific, first, a whole scene representation

$$\theta = F(\mathcal{I}, \mathcal{D}, \mathcal{P}) \tag{1}$$

is needed to serve as the scene knowledge memory for querying the target object because the multi-subscene environment is too big to be perceived through the observation of a single view. Then, a pose estimation method is required to obtain the manipulation target object's pose

$$p = G(t, \theta), \tag{2}$$

where $t$ is the extracted query from $T$. Furthermore, to model the changes in the scene, a scene editing method is needed to update the scene representation

$$\theta' = H(\theta, T). \tag{3}$$

To enable a robotic with the ability to reconstruct and inpaint the scene and apply fine-grained manipulation across subscenes, the following properties are required.

**Understanding of the whole environment and each subscene.** For an unstructured environment with multiple subscenes, prior knowledge of the whole scene and a detailed understanding of each subscene is vital for tasks across subscenes. Given a set of images $\mathcal{I}$ with depths $\mathcal{D}$ and corresponding camera poses $\mathcal{P}$, an entire 3D scene can be reconstructed using 3D Gaussians $\theta$. Depth and pose input are utilized to get a reliable initialization of Gaussians $\theta_{\text{init}}$ with semantics encoded following[3, 27]. Moreover, by incorporating diffusion priors into estimating the depth images, the missing observation regions and regions that are inaccurate from depth camera shots can be completed. The refined depth $\mathcal{D}'$ can be unprojected to a fine-grained point cloud and, together with $\theta_{\text{init}}$, can be fine-tuned to form precise Gaussians $\theta$.

**Imagine before you do it.** For precise manipulation across subscenes, the robotic needs accurate poses of the origin and destination of target objects. Scene inpainting allows for making desired changes to the scene in a manner similar to imagination. For example, in tasks like pick-and-place, by inpainting the scene to remove the object from its original position $p_{\text{ori}}$ and add it to the intended destination $p_{\text{dest}}$, the robotic can acquire a precise pose for picking up and placing the object and obtain updated 3D Gaussians $\theta'$.

**Moving while optimizing.** Cross-subscene manipulation tasks involve both the mobile base movement and robotic arm manipulation. However, optimizing the precise pose of the target in the unstructured scalable scene often takes time. An efficient approach is to navigate the robotic to the vicinity of the target position using a roughly estimated coarse pose $p_c$. During the robotic's movement, a fine-grained pose $p_f$ can be further optimized in a narrowed-down subscene region. This two-step process allows for a more time-efficient execution of the task, during which $p_c$ can be obtained by querying the above-mentioned initial Gaussians $\theta_{\text{init}}$ and the fine-grained pose $p_f$ can be obtained from fine-tuned and inpainted Gaussians $\theta'$.

### 3.2 Multi-subscene Gaussians Reconstruction and Refinement

The effectiveness of using 3D Gaussians as a 3D scene representation heavily relies on the allocation of initial points[51]. To get a reliable initialization $\theta_{\text{init}}$, we unproject the depth maps $\mathcal{D}$ to 3D space with poses $\mathcal{P}$. However, directly using depth captured by cameras may lead to inaccurate values, especially for objects at the visual edges and for reflective or translucent objects. What's more, it's often hard to get sufficient observation of a multi-subscene environment, which leads to missing views of some partials. To address this issue, we employ diffusion-based depth completion to obtain more accurate depth for fine-tuning the initial Gaussians. Fig. 3 shows the visualization results.

The diffusion-based depth completion process takes a set of RGB image $I$, depth image $D$, and an object mask $M$ of the target region as input. The mask $M$ indicates regions that need to be refined. In our approach, Grounded-Light-HQ-SAM[3] is employed as the segmentor to produce $M$. A pre-trained Variational Auto-Encoder(VAE) $\mathcal{E}$ and a U-Net-like[52] denoiser $\mathcal{U}$ are employed

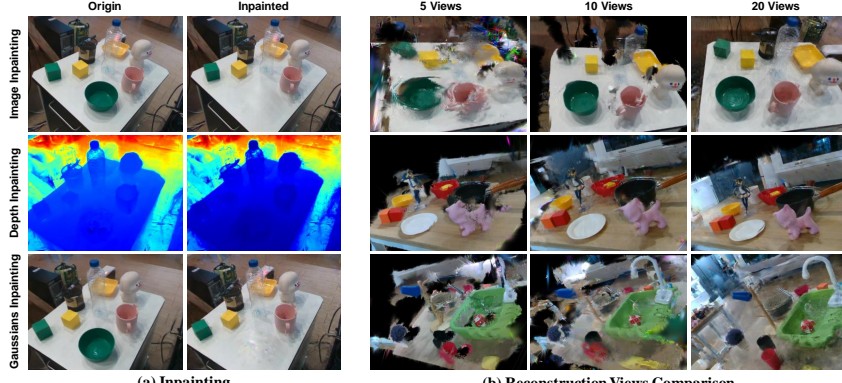

(a) Inpainting      (b) Reconstruction Views Comparison

Figure 4: The inpainting process of Gaussians is shown in (a). The color image is inpainted following SDXL[25], and with the target mask, the depth value in the mask is updated by employing diffusion policies. The Gaussians are updated with inpainted color and depth images consequently. (b) compares the training views of Gaussians for each subscene. It can be seen that only 20 views can reconstruct Gaussians well enough.

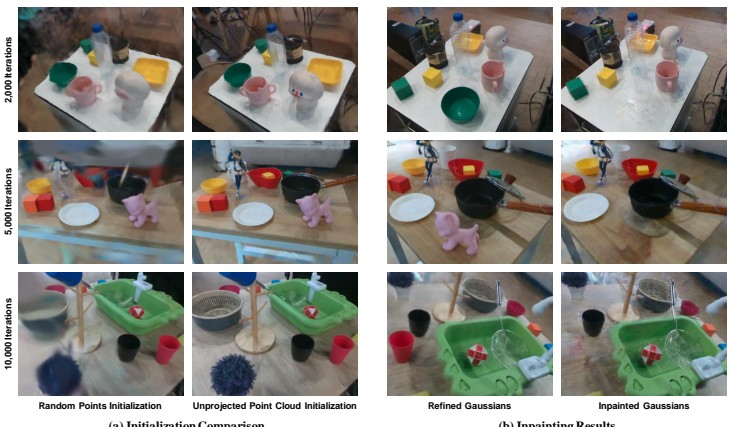

Random Points Initialization   Unprojected Point Cloud Initialization     Refined Gaussians     Inpainted Gaussians

(a) Initialization Comparison      (b) Inpainting Results

Figure 5: (a) compares the initialization method. It can be seen that more accurate Gaussians can be obtained by leveraging fewer optimization iterations with the 3D point cloud initialization than random initialization. (b) shows the inpainting results on the fine-grained Gaussians.

in the diffusion process. Firstly, the depth $D$ is normalized and repeated to form a 3-channel input $D'$, together with $I$, it is encoded by VAE to embeddings $e_{D'}$ and $e_I$. The encoded depth embedding is further noised with $e_{D'}^t = \alpha^t e + \theta^t \epsilon$. The mask $M$ is also down-sampled to match the encoded embeddings as $M'$. Consequently, the noised embeddings at step $t$ can be denoted as

$$e_t = \{e_{D'}^t, e_{D'} \odot M', e_I, M'\}. \tag{4}$$

The loss function of the diffusion process can be denoted as

$$\mathcal{L} = \mathbb{E}_t ||\epsilon - \mathcal{U}(e_t)||_2, \tag{5}$$

where $t \in \{1, 2, \ldots, T\}$ is the diffusion steps. The process is optimized following the DDPM[39] strategy. After completion, the depth $\mathcal{D}'$ is unprojected to 3D, and together with $\theta_{\text{init}}$, is fine-tuned to get the refined Gaussians $\theta$ with several iterations' optimization.

## 3.3 Scene Inpainting for Manipulation Imagination

Precise manipulation in the multi-subscene environment needs fine-grained object pose on the origin and destination. Taking the pick-and-place task as an example, the pose of pick targets can be obtained by direct estimation. However, for the placement target, it's hard to get an accurate pose straightforwardly. To address this, we propose an "imagine and do it" approach. We inpaint the target object into the destination subscene and estimate the pose of the inpainted object as the placement pose. Simultaneously, we erase the object at the origin pose and update the related subscenes.

Specifically, to inpaint the changes in the environment, we select a set of reference views $v_s$ in subscene $s$ for inpainting. Firstly, the depth $D_{v_s}$ can be obtained from the Gaussians $\theta$. The color image $I_{v_s}$ inpainting is processed using an SDXL-based[25] model $\tilde{I}_{v_s} = \text{SDXL}(I_{v_s} \odot M_{v_s})$, and the depth image is completed as mentioned above

$$\tilde{D}_{v_s} = \mathcal{C}(D_{v_s}, \tilde{I}_{v_s}, M_{v_s}), \tag{6}$$

where $\mathcal{C}$ is the depth completion process. The loss function is conducted as

$$\mathcal{L}_{v_s} = \lambda_1 ||I'_{v_s} - \tilde{I}_{v_s}||_1 + \lambda_2 \cdot DSSIM(I'_{v_s} - \tilde{I}_{v_s}), \tag{7}$$

where $I'_{v_s}$ is the rendered RGB and $\lambda_1$ and $\lambda_2$ is the loss weight.

### 3.4 Coarse-to-fine Pose Refinement for Mobile Manipulation

**Scene Knowledge Prior.** In environments with multiple subscenes, the mobile base needs to move back and forth within the scene to collaborate with the robotic arm in completing tasks. However, optimizing the precise pose for robotic arm operations in the whole scene can be time-consuming. It would be highly inefficient to wait for all optimizations done before taking any movement. As mentioned earlier, a rough pose can be obtained by querying the initial Gaussians $\theta_{\text{init}}$ for the mobile base to approach the target within the reachable range of the robotic arm. Moreover, based on the retrieved pose, the range of optimization can be narrowed down to the relevant subscene.

**Depth Completion Empowered Fine-grained Pose Estimation.** Depth values obtained from depth cameras can be inaccurate for pose estimation, especially at the camera's edges or in regions with translucent or reflective objects. Additionally, there may be missing or insufficient observations in certain partials during the entire scene capture process. The depth-completed Gaussians $\theta$ can be projected onto the queried views from coarse poses for more precise estimation, as shown in Fig. 3.

## 4 Experiments

| Views | Render and Compare | | Inpainting | |
|---|---|---|---|---|
| | Deviation | Speed | Deviation | Speed |
| View$_1$ | 1.26 | 10.8 | 0.72 | 18.2 |
| View$_2$ | 1.55 | 11.2 | 0.80 | 17.7 |
| View$_3$ | 1.47 | 10.9 | 0.68 | 18.1 |

Table 1: Comparison of the render-and-compare and inpainting method on the scene editing task. The average pose Deviation is evaluated in *cm* and optimization speed is evaluated in *Hz* of each subscene. View 1~ 3 are sampled from 3 subscenes.

| Methods | Pick & Place | Stack | Pouring | Tidy-up |
|---|---|---|---|---|
| F3rm[2] | 5/10 | 2/10 | 0/10 | 1/10 |
| HomeRobot[6] | 5/10 | 4/10 | 1/10 | 2/10 |
| **Ours\*** | 8/10 | 5/10 | 3/10 | 5/10 |
| **Ours** | **9/10** | **7/10** | **5/10** | **7/10** |

Table 2: Comparison of success rate on tasks across-subscene. Evaluation is conducted on various challenging tasks. To enable F3rm[2] and HomeRobot[6] with cross-subscene ability, we fine-tuned them to reconstruct the subscenes in an aligned space for given tasks. 'Ours*' stands for using Gaussians not refined by completed depth.

**Experiments Setups** We collect RGB-D images with a RealSense D435 camera mounted on a Franka Panda arm with a movement base. For each subscene tabletop, we scan 20 images with a resolution of $640 \times 480$, and we set three tables at a distance from each other as multiple subscenes in our environment. Additionally, we take $2 \sim 3$ images of the whole scene far from three tables to get a good visualization. The scans are set surrounding the tabletop without a fixed trajectory. Our models and real-world robotic experiments are trained and run on an Nvidia RTX 3090 GPU.

**Multi-subscene Reconstruction** We conduct experiments comparing Gaussian initialization, showing that reliable coarse Gaussians can be obtained with the unprojected 3D point cloud initialization. Fig. 5 shows the comparisons. On the other hand, the reconstruction of an unstructured scalable scene typically demands extensive observations. However, our coarse-to-fine approach, coupled with the diffusion policy, significantly reduces the required number of sample views. The comparative results depicted in Fig. 4 demonstrate that even with just a few input views, each subscene can be reconstructed well.

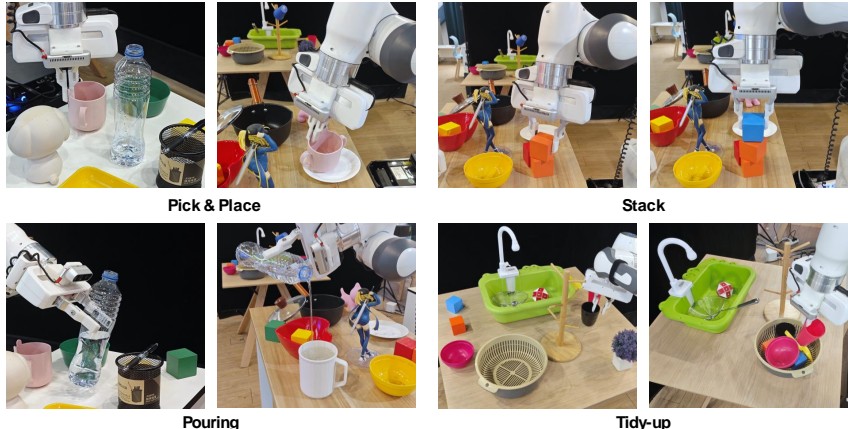

Figure 6: Examples of cross-subscene tasks which can be applied by our proposed method. We demonstrate the ability to process tasks like stacking, pouring, and rearranging.

**Target Driven Scene Inpainting**. For manipulation across subscenes, estimating the target object's origin and destination position is of great significance. To demonstrate the effectiveness of inpainting, we compare it with a straightforward approach that directly optimizes the 3D Gaussians and renders the resulting RGB image to compute a rendering loss. The results are shown in Tab. 1. It can be seen that with Gaussians inpainting in the 3D space, the pose deviation and optimization time consumption are less. Fig. 5 shows some inpainting examples on 3D Gaussians.

**Depth Completion Empowered Pose Estimation** Depth completion is an effective approach to get a more dense and accurate pose estimation of the target object. To evaluate the depth completion, we compare the depth map as well as the success rates of challenging tasks before and after depth completion. Results in Fig. 3 and Tab. 2 show that depth completion plays a vital role when the raw depth input is inaccurate, especially on translucent or reflective objects.

**Fine-grained Manipulation Evaluation**. To evaluate our proposed compositional target-driven mobile manipulation pipeline, we evaluate the manipulation results on tasks spanning across subscenes. As for the comparing methods, we realize the state-of-the-art manipulation method for scalable environments and regard the whole scene as the workspace. The success rate on various challenging tasks is evaluated as the metric. Results shown in Tab. 2 and Fig. 6 demonstrate the advantages of our proposed method.

## 5 Conclusion

This paper proposes *TaMMa*, which leverages cross-subscene fine-grained mobile base movement and robotic arm manipulation efficiently and effectively. The 3D point cloud, unprojected from depth images and poses, serves as a reliable 3D Gaussians initialization to provide a rough pose estimation of the target and narrow down the further optimizing scope of the target pose. Depth completion is employed for precise pose optimization aiming to solve the inadequate observation and inaccurate depth estimation. The completed point cloud is employed to refine the 3D Gaussians and inpaint the scene changes for target manipulation. *Moving while optimizing* and *imagine before you do it* manners are proposed as efficient coarse-to-fine and accurate manipulation pose estimation approaches. Various experiments were conducted on a real robotic to evaluate our proposed method.

**Limitations and Future Work**. Our work have raised questions about two limitations: **I.** Single-view depth completion is not so robust, which hinders optimization in depth-only pose estimation. We plan to explore multi-view knowledge integration to improve depth completion. **II.** Inpainting 3D Gaussians works well for flat objects but struggles with complex, three-dimensional ones. Extracting and re-editing complex objects may not yield satisfactory results. Future research will aim to enhance inpainting techniques for complex geometries.

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

# Appendix

## A  Setups

In this section, we illustrate the physical setup of the environment, robotic, and the target objects. Moreover, we list the parameters and settings of the employed models in detail.

### A.1  Physical Setup

**Robotic Setup.** The construction of mobile base is shown in Fig.A1. The mobile base mainly carries a Franka Panda arm and a PC which takes control of the mobile base and arm. The maximum velocity of mobile base is set to $1m/s$ and the Franka arm is controlled by MoveIt library. The linear velocity is set to $0.2m/s$, and gripper force is set to $5N$. The robotic is placed initially about $1m$ away from the main experimental environment. A Realsense D435 camera is employed and attached to the robotic arm end effector and calibrated with the easy-hand-eye package. The gripper is a 3D-printed model with a length of 5 $cm$. For the mobile base, we employ the SLAMTEC Hermes, equipped with a laser radar for simplified mapping, localization, obstacle avoidance, and navigation. A comparison of the robotic setup with other related works is listed in Tab. 3 which shows our versatility and simplicity.

| Methods | Camera | Robotic Arm | Robotic Base |
|---|---|---|---|
| CLIP-Fields[41] | iPhone 13Pro with LiDAR | Hello Robot Stretch (not used) | Hello Robot Stretch |
| F3RM[2] | RealSense D415 | Franka Panda Arm | - |
| OK-Robot[7] | RealSense D435 | Hello Robot Stretch | Hello Robot Stretch |
| Mobile ALOHA[49] | Logitech C922x Pro Stream Webcam * 4 | ViperX 300 Robot Arm 6DOF * 2 WidowX 250 Robot Arm 6DOF * 2 | Self-constructed |
| AMM.[43] | RealSense D435 RealSense T265 | xArm | AgileX Ranger Mini 2 |
| TeleMoMa[48] | RealSense | Tiago++ robot | Tiago++ robot |
| TaMMa(Ours) | RealSense D435 | Franka Panda Arm | SLAMTEC Hermes |

Table 3: Comparison of robotic setups with representative previous works.

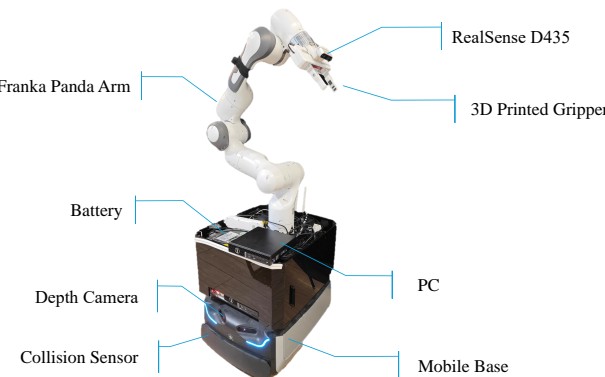

Figure A1: The construction of the robotic setup. The mobile base mainly carries a Franka Panda arm and a PC which takes control of the mobile base and arm.

**Environment Setup.** The subscenes taken as an experiment example in our paper is a multi-tabletop scene in a big room. We set up 3 tables of different sizes in the scene, arranged in a triangular pattern,

with an average distance of 1 $m$ between the tables, as shown in the diagram. The objects on the tables can be categorized as (1) daily-used containers, such as cups, bottles, plates, baskets, pen holders, and a sink, (2) geometric objects like cubes, (3) challenging objects for grasping, such as rubber toys, irregular toy models, and transparent objects. The arrangement of objects on the tables is relatively random.

## A.2   Experiment Setup

As for task evaluation:

**1) Pick & Place.** The task is thought to be completed when robotic successfully catch the target object from a subscene, move to another subscene and put down it on the destination point without falling on the first attempt.

**2) Stacking.** The task is thought to be successful when robotic collects objects from different scenes and stacks them as instructed on the first attempt, during which no object falling happens and the stacked objects won't fall over without external forces.

**3) Tidy-up.** The task is successful when robotic recognizes all the objects needed to be rearranged and carries them to the target destination without leftover.

**4) Pouring.** We prepare $300ml$ water in a plastic bottle. We consider the task successful if robotic could carry the bottle from one subscene to another subscene without falling and pour the water into the target cup more than $150ml$.

## A.3   Model Setup

**Gaussian Reconstruction.** For the Gaussian-Splatting-based scene reconstruction, we employ the widely-used open-source 3DGS[**?** ] code base. The initial point clouds used as the Gaussian initialization are downsampled by a factor of 5. The rendering process generates results as the origin image resolution of $640 \times 480$. Subsequently, Gaussians are optimized for 30,000 iterations across all scenes, utilizing the same loss function, Gaussian density, schedule, and hyperparameters as specified in the original implementation, which includes a learning rate dynamically adjusts based on the iteration number and a densification process that starts early in training and continues until the specified iteration threshold. The densification parameters are finely tuned, with densification performed every 100 iterations, an opacity reset interval of 1,000 iterations, and a gradient threshold for densification set to 0.01.

**Depth Completion.** The depth inpainting process is powered by a diffusion model which is built upon Latent Diffusion Models (LDMs) [24, 39, 25] using a pre-trained Variational Auto-Encoder (VAE) and a U-Net-based[52] denoising architecture. Utilizing the frozen VAE, we encode both the color image and the depth map into a latent space, forming the basis for training our conditional denoiser. The depth map is repeated at channels to form a tri-channel input as an RGB image and is normalized. A composite feature map is constructed by concatenating the encoded depth and image elements. The denoising step is set to 20 at inference as default to trade off the time consumption and the effect. The U-Net-based denoising architecture iteratively refines the depth latent by predicting and removing noise at each timestep, which is managed by the DDIM[39] scheduler to ensure that noise is progressively reduced in control.

**Gaussian Merging and Fine-tuning.** With the corresponding camera pose and obtained depth map of the inpainted image, the 2D inpainted data is unprojected into a 3D colored point cloud from image space. Then, features from the original and inpainted Gaussian point clouds are merged by concatenating their poses, features, and opacities. To remove floaters at the edges of the mask, the minimum number of points within a radius of 0.1 for a point to be considered not an outlier is set to 100 as default. The following fine-tuning process optimizes the model using a combination of L1 loss and D-SSIM (Differentiable Structural Similarity Index) to ensure that the final rendered results closely match the inpainted reference images. The weight parameters are set to 0.8 and 0.2 respectively, reflecting the emphasis on maintaining a balance between pixel-wise accuracy and

perceptual similarity. The optimization is performed over 150 iterations to achieve the final Gaussian model.

# B    Task Videos

We provide videos illustrating our cross-subscene fine-grained manipulation ability on 4 tasks. Each of the interacting objects can be located in the arbitrary sub-scenes.

**Pick and Place.** The input query is in the form of "Move the [A] to the [B]", where [A] and [B] are objects in the whole scene. For example, in the video, we set "Move the pink cup to the white plate", in which objects lie on 2 separate tables.

**Stacking.** The input query is in the form of "Stack the [A] onto the [B]", where [A] are objects to be stacked and [B] is the target place. In the video, we show the result of applying "Stack the orange cubes onto the blue cube". In this case, orange cubes on different tables will be collected and placed on the blue one.

**Pouring.** The query is in the form of "Pour the liquid in the [A] into the [B]", where [A] is the container that has liquid in it and [B] is the target container. We show the result of "Pour the liquid in the bottle into the white cup". In this example, the robotic carries the bottle smoothly to the cup and rotates the bottle to pour.

**Tidy-up.** The query is in the form of "Tidy up the table with [A]", where [A] is the representative objects of a subscene. For example, the video shows the result of "Tidy up the table with toys and cups". In this example, the small objects on the table with toys and cups will be rearranged into a basket.

**Long term tasks in complex environment with obstacles and more than 3 subscenes.** To demonstrate the manageability of our approach in a complex environment with multi-subscenes beyond toy-example tabletops, we conduct an experiment instructed by "Pour the water from the shelf bottle into the pot, discard the bottle, and place the cleaning sponge in the pot." The environment is conducted with 4 subscenes: shelf, sink, table1, and table2, including obstacles on the way. The subscene setting is just like the daily look as shown in Fig.B1.

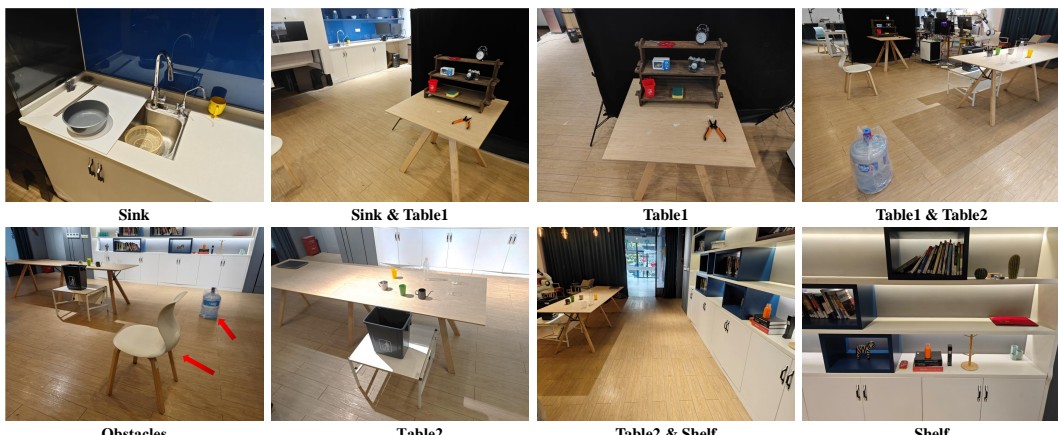

Figure B1: The complex environment setup with obstacles and more than 3 subscenes, which is conducted to demonstrate the manageability of our proposed method.

# C    Mobile Manipulation

**Point Cloud Extraction** To extract a scene-wide point cloud for manipulation, we reconstructed the entire scene based on 3D Gaussian Splatting [**?** ] and performed depth completion using diffu-

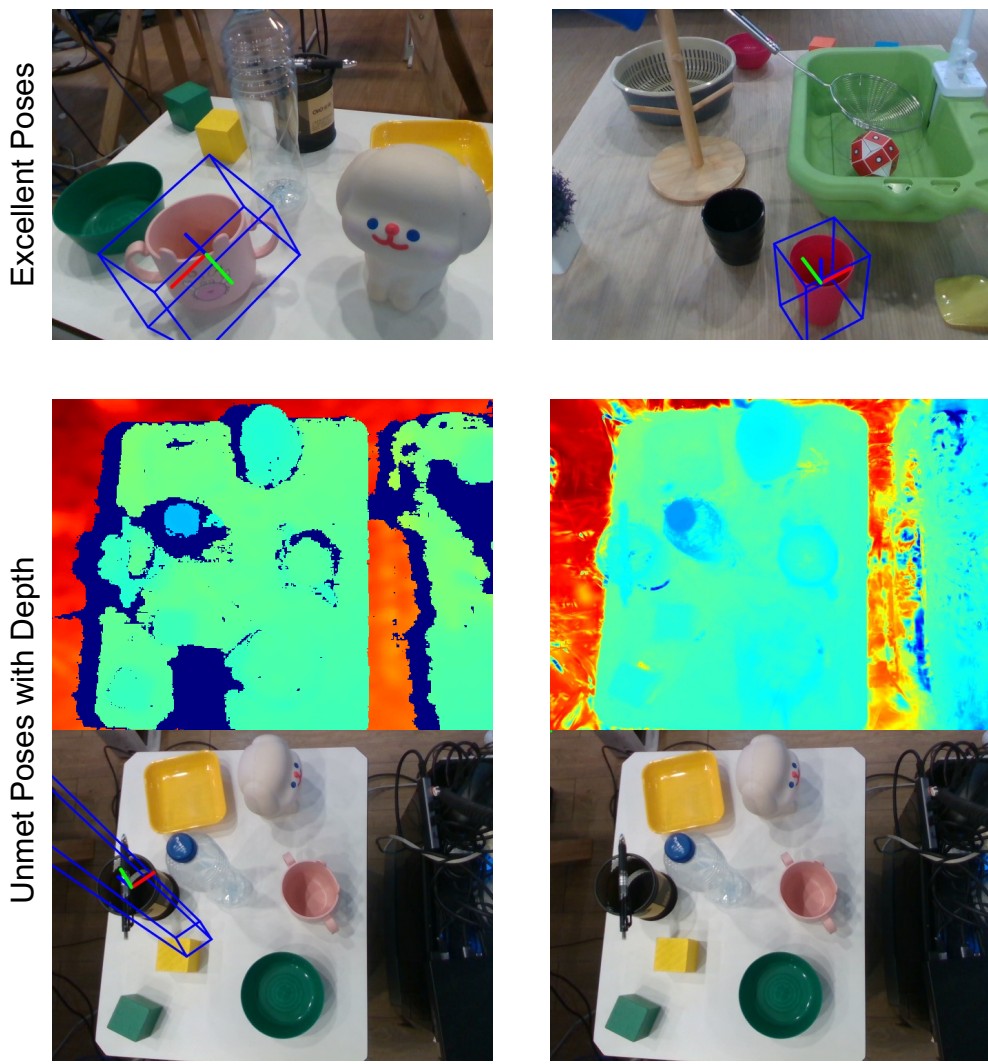

Figure C1: The qualitative results of pose estimation under different scenes. The top row shows the excellent poses under the corresponding views. The two bottom rows show unmet poses ("Pen") with depth. In these bottom rows, the left column presents the original depths acquired by the depth camera, while the right column shows the depths completed using our method.

sions [39]. Starting from the completed depth map, we performed back-projection to obtain point clouds of several sub-scenes. Subsequently, we transformed the point clouds into the robot's working coordinate system , and executed specific navigation or manipulation tasks based on the transformation matrices of the movement base and end-effector relative to the working coordinate system.

**Navigation and Manipulation** The navigation process is finished by employing the API provided by the SLAMTEC mobile base. Specifically, a 2D topdown occupancy map is built for the environment as the scene map. Based on the occupancy map, the trajectory planning and obstacles avoidance is completed by querying the map with the target position and the current position. By generating a set of waypoints, the mobile base is guided to the target position. As for manipulation, Franka Panda Arm is operated through the MoveIt! library [**?** ]. We provide the 6 DoF pose of the target and employ the movement API to approach and operate the gripper to close and open depending on the width of the estimated target objects.

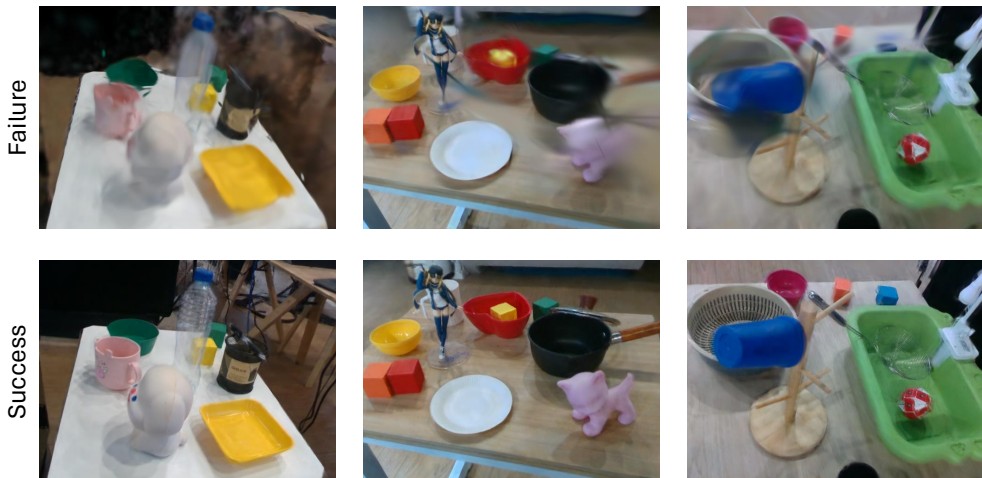

Figure D1: The comparison of reconstruction failure caused by camera jittering and the successful results.

**Motion Planning based on 6D Pose** Some fine-grained robotic manipulations mainly rely on the 6D pose of the target for motion planning [**?** ]. In the motion planning process of TaMMa, we integrated a depth-only pose estimation method, as the state of objects in 3D space is crucial for determining meaningful manipulation points across various tasks. Specifically, since the poses of intra-category instances are pre-normalized [**?** ], it becomes highly advantageous to define task-relevant category-level grasping poses. These defined grasping poses are then transformed from object coordinates to camera coordinates using the estimated 6D object poses. The robot subsequently performs motion planning to move its gripper to the target grasping pose to accomplish each task. Based on the poses of different category-level objects and the tasks to be performed (e.g., pouring, stack), we utilized MoveIt! [**?** ] to set up target-driven action sequences. Therefore, the success rate of robotic manipulation heavily depends on the accuracy of pose estimation. Fig. C1 presents the qualitative results of pose estimation across different scenes. The top row of Fig. C1 demonstrates the excellent poses in most cases. However, there are instances where the results are suboptimal, often due to inaccurate depth data. This can be observed in the bottom two rows. When we attempt to acquire the pose of the "black pen," the original depth (left column) and the completed depth (right column) are both unsatisfactory, affecting the subsequent pose estimation process.

## D  Failure Cases

**Capture Failures.** Accurate camera pose is crucial for scene reconstruction. During the experimentation process, we discovered that the jittering and offset errors of the mobile base and robotic arm could result in inaccurate camera poses used for scene reconstruction. Consequently, this leads to errors in the initialization and optimization processes of Gaussians, resulting in issues such as ghosting in the reconstructed scene. One possible solution is to reduce the movement speed of the base and robotic arm, capture data only when they are stable, and minimize frequent movements of the chassis as much as possible. The visualization of comparison is shown in Fig. D1.

**Segmentation Failures.** Our experiments have shown that when using a segmentation model based on SAM[3] as a mask for image inpainting and depth completion, the inaccuracy of the mask can result in incomplete object editing, leaving behind edges or blurry traces. A more precise mask can achieve better visual effects. The comparison inpainting results between using inaccurate masks and refined masks are shown in Fig. D2.

Origin Gaussians   Gaussians Inpaiting Failure   Gaussians Inpaiting Success

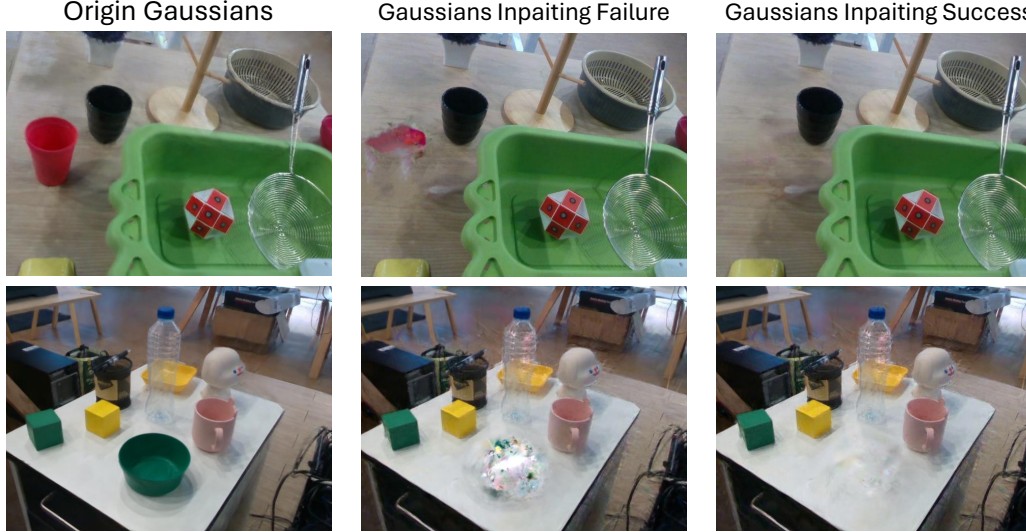

Figure D2: The comparison of inpainting failures caused by inaccurate masks and the successfully inpainted Gaussians results.

## E   Detailed Illustration of Limitations

**Monocular Depth Estimation.** In practical applications of robotic arm manipulation, it is common to use the depth values directly captured by a depth camera as the input for scene modeling and object pose estimation. This is because the depth values provided by a depth camera are absolute depth values, which differ from the relative depth typically obtained through monocular depth estimation. This absolute depth information is highly valuable for practical operations. However, depth camera captured results often exhibit errors and significant uncertainty, particularly at the edges of the camera's field of view and when dealing with transparent or reflective objects. These factors raise issues in performing operations in challenging scenarios. In this paper, we utilize a diffusion-based depth completion method to partially address the discontinuities and errors present in the depth camera results. However, the completed depth results still contain certain errors compared to the original absolute depth input. How to leverage depth maps from multiple viewpoints to supplement depth maps from a single viewpoint and obtain a dense and reliable point cloud remains an open question. What's more, the object occlusion may also cause depth completion failure. As mentioned above, even the diffusion-based method is also unable to solve the depth prediction problem for objects with severe occlusion. Additionally, for transparent objects, objects located behind them can also affect depth and pose estimation. To manage this issue in 3D space, amodal-based method will be our future work direction.

**3D Objects Inpainting.** Calculating precise poses for robotic arm manipulation has always been an open and challenging problem, especially in scenarios lacking constraints. For example, it is difficult for a robotic arm to place a cup in the exact middle of the plate to achieve the same level of precision as a human. Similarly, stacking blocks together is challenging for a robotic arm to achieve a perfect appearance. These results stem from two main reasons: the difficulty of endowing the robotic arm with real-time adjustment capabilities and the inability of the arm to obtain accurate pose information of the target location. Through a comparison with existing successful robotic arm grasping tasks, we observed that while the estimation of graspable object poses can now yield fairly accurate results, it remains challenging to estimate the precise poses required for tasks such as placing a cup in the center of a plate or achieving perfect block stacking. Consequently, we propose using inpainting techniques, employing a "think before you do it" approach, to address this issue. We first edit the images and depth maps, followed by editing Gaussians to create a 3D representation of the desired task-completed scenario. We then estimate the poses of the target objects in this scene

to obtain a more accurate target pose for interaction. However, implementing this approach still presents difficulties. The current Gaussians inpainting methods perform well for editing relatively planar objects but struggle with editing voluminous and complex-shaped objects. Therefore, our future research direction will focuse on how to edit objects in 3D space to obtain accurate results that can be used for interactive operations.

**More Complex and Larger-scale Scenes.** We have successfully evaluated our approach in an environment at about $200m^2$ with about 10 subscenes. The obstacles set on the way are boxes, chairs, and objects with regular geometry with widths larger than $5cm$. Each obstacle is observed by about 3 frames from the camera when recording the environment. As for environments with a larger number of subscenes, the whole scene is represented with united Gaussians, and subscenes are divided by humans according to the distribution of optional interactive objects. We consider a part as a subscene when there are many objects that need careful perception. The number of subscenes doesn't affect the Gaussians optimization. However, more sampled images are needed for better results when there are more objects. As for the movement and manipulation, the target object localization takes a text query as input and outputs a coarse 3D position, while manipulation takes a task query as input and outputs manipulation strategy. The above steps are totally free from the number of subscenes. Consequently, the number of subscenes does not affect our approach. As for environments with larger scales and areas, this mainly affects the computation and memory that build the Gaussians of the environment. Taking advantage of our coarse-to-fine approach, the point cloud initialization of the Gaussians could be sub-sampled to an appropriate scale, and further optimization could be conducted on the subscene with more interests. Further to this end, lots of recent works have made great progress on expanding the Gaussians application to un-constrained and larger scenes. This would be one of our future directions to optimize the Gaussians for application in a larger-scale scene.

**Transparent, Reflective, and Complex Shape Objects.** The challenging objects' appearance mainly affects the depth estimation and the inpainting process. Reflective and transparent objects hesitate in their optical characteristics, making it difficult for depth cameras to capture precise depths, thus affecting subsequent depth completion processes. Transparent objects are easily influenced by other objects when occluding or being occluded, causing depth estimation to be affected, resulting in phenomena like depth truncation. Similarly, in complex backgrounds, similar phenomena may occur. Reflective objects are prone to distortion and pseudo-shadow phenomena in depth. Objects with complex shapes affect the accuracy of depth estimation, with depth cameras showing differences in attention to detail, and small parts being easily overlooked during depth estimation. Furthermore, for object inpainting, errors in depth estimation and the complex relationship between shapes, textures, and backgrounds can lead to a decrease in inpainting effects. Fig.E4 and E3 show some examples of objects with transparent ones and complex shapes, demonstrating our model's ability to manage a number of challenging objects. However, such challenging situations can cause blurs, shadows, and noises. Therefore, one of our future research directions is to optimize the depth and pose estimation of challenging objects in 3D space under multi-view observations and introduce various shape priors for common shapes to further enhance the representation accuracy of objects.

## F   Implementation Details of Comparison Methods

For the cross-subscene mobile manipulation task, only a few works have been open-sourced. In this paper, we choose F3rm[2] and HomeRobot[6] as the comparison methods. To enable them with the cross-subscene ability in our environment, we re-implement and fine-tune these methods for the aiming tasks. Regarding real-world robotic tasks, we conduct evaluations across ten distinct scenes per task, each featuring novel objects from various categories in arbitrary poses. We ensure that the scenes are reset to approximately the same configuration for each method being compared.

**F3rm.** We implement F3rm to expand its workspace to the cross-subscene environment. For the data source, the same images from RealSense D435 are used as input, and to be fair, the depth images are also employed to train the implicit representation as an additional loss. The camera poses come

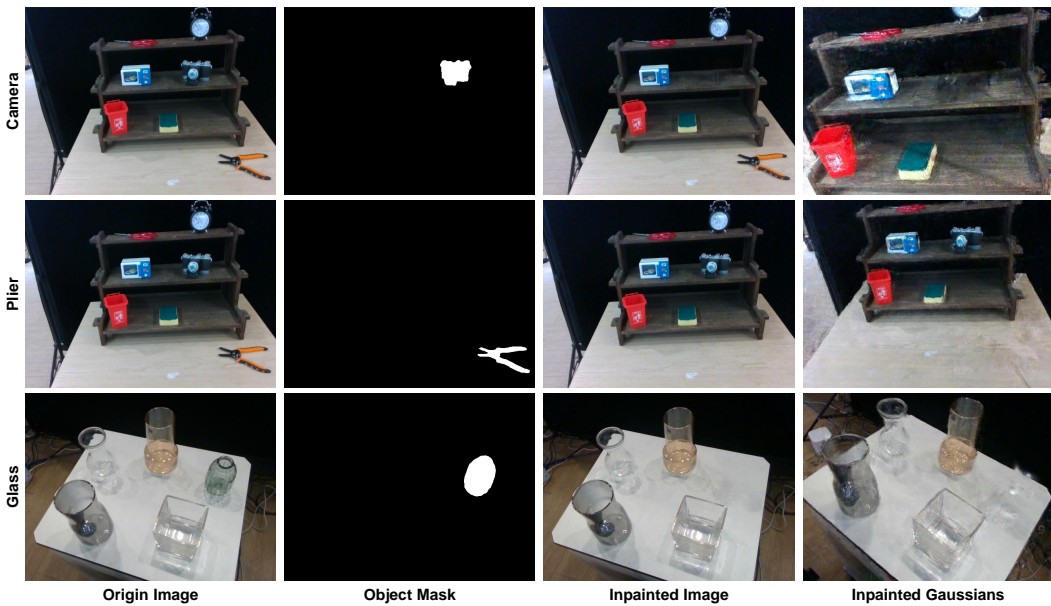

**Figure E3:** The example of inpainting transparent objects and objects with complex shapes. Current model can handle a number of these challenging objects but some blurs, shadows, and noise still exist.

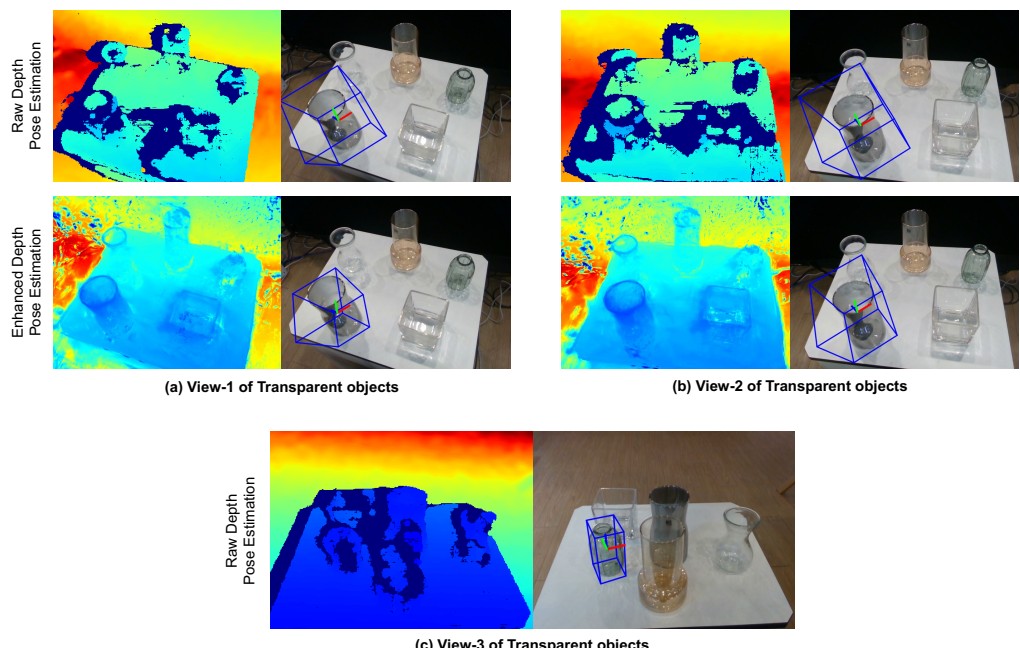

**Figure E4:** The qualitative results of transparent objects from multiple views. (a) and (b) illustrate the pose estimation of the same object ("Transparent Bottle") from different viewpoints. The top trow presents the results based on the original depth, while the bottom row shows the results based on the enhanced depth. (c) shows the different bottle from another view, where a satisfactory pose estimation result is achieved even with the raw depth.

from the calibrated camera on the robotic arm. As the environment expands, the need for VRAM increases obviously. Fairly, we maximize the VRAM usage of the RTX 3090 by employing a smaller feature resolution, as mentioned in F3rm, and ignoring the regions out of the tabletop workspace. As for the manipulation process, we first drive the mobile base to ensure the manipulation targets are reachable for the robotic arm and optimize the target pose for manipulation.

**HomeRobot.** We implement HomeRobot in our environment for the cross-subscene task. The provided Detic and Grounded-SAM libraries are employed to get semantics. The exploration of the environment is replaced by feeding the recorded RGB-D sequences to the HomeRobot, and the navigation and planning of the mobile base is implemented in the same way as our proposed *TaMMa*. For mobile manipulation, the mobile base is first guided to approach the target, ensuring the targets are reachable, and the Contact Graspnet is employed to generate the 6 DoF grasp pose of the target. The pose of the receptacle, the location on which the object is placed, comes from adjusting from a reference object. For example, to put the cup on a plate, we optimize the pose of the plate and add a bias on the y-axis to get the target pose of the cup.

## G    Supplementary Pseudocode

**Diffusion based Depth Completion.** Algorithm 1 outlines the process for diffusion based depth completion, which provides a structured approach to filling in missing depth information with multiple views of RGB-D inputs, addressing common challenges in depth sensing tasks such as incomplete data or occlusions. Utilizing a diffusion process that leverages latent diffusion models (LDMs), the algorithm mainly consists of three steps: Initialize Diffusion Process, Perform Depth Diffusion, Refine Depth Maps. Finally, the algorithm outputs the refined and completed depth maps, which can be used for the following 6D Pose Estimation task to enhance the effect of its depth-based estimation method.

---

**Algorithm 1** Diffusion based Depth Completion

---

1:  **Input:**
2:      $I_{RGB}$: Set of RGB images of sub-scenes
3:      $D_{init}$: Set of initial depth maps
4:      $M$: Set of masks delineating the target completion region
5:  **Output:**
6:      $D_{enh}$: Set of completed depth maps
7:  **Step 1: Initialize Diffusion Process**
8:  **for** each $d_j \in D_{init}$ and $i_j \in I_{RGB}$ and $m_j \in M$ **do**
9:          Apply a linear normalization via an affine transformation to $d_j$
10:         Leverage the frozen VAE encoder to encode normalized depth $d_j'$ and $i_j$ into latent space, which produces a 4-channel feature map $z_j$
11:         Resize $m_j$ to align with the dimensions of $z_j$, yielding downsampled $m_j'$
12:         Create a composite feature map $z_t$ by applying random noise to the above latent code
13:  **end for**
14:  **Step 2: Perform Depth Diffusion**
15:  **for** each $z_t$ **do**
16:      **for** $t = 1$ to $T$ **do**
17:              Initialize U-Net with trained weights for depth completion and feed $z_t$ into the U-Net-based denoiser
18:              Denoise and complete the depth map with interpolated depth values in uncertain areas
19:      **end for**
20:          Derive $d_{enh}^j$ from the latent representation decoded by the VAE decoder
21:  **end for**
22:  **Step 3: Refine Depth Maps**
23:  **for** each $d_{enh} \in D_{enh}$ **do**
24:          Apply channel-wise averaging to smooth the depth map $d_{enh}$ for post-processing
25:  **end for**
26:  **Return** $D_{enh}$

---

**Fine-grained 6D Pose Estimation based Manipulation.** Algorithm 2 outlines the process for fine-grained 6D pose estimation based manipulation, which aims to precisely estimate the pose and size of objects within a scene and subsequently generate appropriate action sequences for cross-subscene manipulation tasks. The algorithm mainly consists of four primary steps: Grounded Visual Perception, Depth-based Category-Level Pose Estimation, Coordinate Transformation, Generate Target-driven Action Sequences. Finally, the algorithm returns the estimated 6D poses, 3D sizes, and the generated action sequences, which collectively enable precise and efficient manipulation across multi-subscenes.

---

**Algorithm 2** Fine-grained 6D Pose Estimation based Manipulation

---

1: **Input:**
2:    $I_{RGB}$: Set of RGB images of sub-scenes
3:    $D_{enh}$: Set of depth-enhanced depth maps
4:    $Q$: Set of attributes of instances as queries
5:    $T$: Specific types of manipulation tasks
6: **Output:**
7:    $P_{6D}$: Estimated 6D pose for each instance
8:    $S_{3D}$: Estimated 3D size for each instance
9:    $A$: Target-driven action sequences based on estimated poses
10: **Step 1: Grounded Visual Perception**
11: **for** each $q \in Q$ **do**
12:    **for** each $I \in I_{RGB}$ **do**
13:       Apply Grounded-Light-HQSAM to obtain mask $M_{I,q}$ for query $q$ from $I$
14:    **end for**
15: **end for**
16: **Step 2: Depth-based Category-Level Pose Estimation**
17: **for** each $M_{I,q}$ and $D \in D_{enh}$ **do**
18:    Back-project the depth map $D$ and mask $M_{I,q}$ into point clouds in the observed view
19:    Utilize MVPoseNet6D to estimate 6D pose $P_{6D}^i$ and 3D size $S_{3D}^i$ for each instance $i$ from the point clouds
20: **end for**
21: **Step 3: Coordinate Transformation**
22: **for** each $P_{6D}^i$ **do**
23:    Transform pose $P_{6D}^i$ from the object coordinates to the camera coordinates
24:    Transform pose $P_{6D}^i$ from the camera coordinates to the robotic arm base coordinates
25: **end for**
26: **Step 4: Generate Target-driven Action Sequences**
27: **for** each $P_{6D}^i$ **do**
28:    Based on manipulation task type $T$, integrate each target pose $P_{6D}^i$
29:    Invoke MoveIt! to generate the action sequence $A_i$
30: **end for**
31: **Return** $P_{6D}, S_{3D}, A$

---

