# OpenReview forum: "TaMMa: Target-driven Multi-subscene Mobile Manipulation"
_robot-learning.org/CoRL/2024/Conference — CoRL 2024_

### Official Review · Reviewer_GbuM · 2024-07-18
**TaMMa Review**

**Originality:** 4
**Technical Quality:** 4
**Clarity Of Presentation:** 4
**Potential Impact:** 3
**Recommendation:** 4
**Confidence:** 4

**Review:**

Quality:

This method makes comparisons to F3rm and HomeRobot. It is challenging to find and compare to other methods for complex robotic system performance, I appreciate the effort that went into benchmarking this approach against prior works.


Clarity:

Overall the paper and video present clearly.

For this video it would be nice to include additional examples of image / depth inpainting rather than just videos of the robot moving & doing tasks.

More information about Diffusion based depth completion described in section 3.2 would be helpful. I am not confident that I understand how this is working from reading your paper. Adding some pseudo code to the appendix or an additional figure describing how this is formulated would be really helpful. My understanding is that you take an original depth image, add noise to it, and then train the diffusion-denoiser based on its ability to remove this added noise. If I am understanding this correctly, it isn’t clear to me that the training of this denoiser would actually work at removing noise artifacts common to realsense cameras (transparencies, reflective surfaces, object edges where stereo matches are challenging to obtain, etc). Some additional description around this section or in the appendix would be greatly appreciated.

Alternatively additional clarity could be provided by releasing some of the associated code.

Originality:

This work leverages rough 3D gaussians for rough pose iestimation, that are then further refined. The presented work does a great job at demonstrating how 3D gaussians, SAM, and multi-view depth images can be integrated to enable a mobile manipulator.

Significance:

I think this work presents a strong baseline for mobile manipulation.

**Quality Of The Limitations Section:**

3

**Questions For Rebuttal:**

More information about Diffusion based depth completion described in section 3.2 would be very helpful.

**Robotics Focus:**

4

**Summary Of Paper:**

This work presents TaMMa (Target-driven multi-subscene mobile manipulation). A franka arm with a realsense camera on a mobile base accomplishes navigation + manipulation tasks across 3 different nearby tables. The approach leverages 1) Diffusion based depth completion, 2) 3D gaussians for scene representation, and 3) Scene inpainting to visualize what the effects of pick & place like actions on the environment.   TaMMa first scans only a few posed RGB-D frames of each subscene and unprojects them to a sparse 3D point cloud to initialize 3D Gaussians. Grounded-Light-HQ-SAM[3] extracts queries and encodes semantics into Gaussians, which can be queried to provide a coarse pose of the manipulation target.

**Summary Of Recommendation:**

I think this work presents a strong baseline for mobile manipulation, and a nice synthesis of SOTA methods to accomplish robotic tasks on real hardware.

---

### Official Review · Reviewer_j6RG · 2024-07-23

**Originality:** 3
**Technical Quality:** 3
**Clarity Of Presentation:** 3
**Potential Impact:** 2
**Recommendation:** 3
**Confidence:** 3

**Review:**

### Summary:
The paper proposes TaMMa, Target-driven Multi-subscene Mobile Manipulation, a method for target-driven multi-subscene mobile manipulation using 3D Gaussian splatting, depth completion, and scene inpainting. It enables robots to perform manipulation tasks across multiple subscenes efficiently.

### Strengths:
- This paper presents a method to integrate of 3D Gaussian splatting, depth completion, and scene inpainting.
- The multi-subgoal strategy presented in the paper is a practical solution to improve target-driven navigation.
- The paper demonstrates that robots has the ability to handle complex tasks like pouring and stacking across scenes.
- The paper provides a relatively thorough experimental evaluation, comparing the proposed method with existing approaches.

### Weaknesses:
- Limited evaluation on only one robotic platform (Franka Panda arm with mobile base).
- Lack of comparison with some recent state-of-the-art mobile manipulation methods, though admittedly there are few that are open-sourced or easily reproducible
- The In-painting method is limited for not complex 3D objects, as noted by authors. The scalability of the proposed approach to more complex and larger environments is not very well discussed.

**Quality Of The Limitations Section:**

3

**Questions For Rebuttal:**

- How does the method scale to environments with more than 3 subscenes?
- How sensitive is the method to errors in initial scene reconstruction or depth estimation?
- If attempting to transfer to a different robot configuration, how will the approach be on different robot platforms or arm configurations?

**Robotics Focus:**

4

**Summary Of Paper:**

The paper presents TaMMa (Target-driven Multi-subscene Mobile Manipulation), a method designed to enable a mobile base and a arm to efficiently navigate and perform single-arm manipulations across multiple similar scenes.

**Summary Of Recommendation:**

-

---

### Official Review · Reviewer_pWYS · 2024-07-27
**Separation of navigation and manipulation stands out as a critical weakness**

**Originality:** 4
**Technical Quality:** 4
**Clarity Of Presentation:** 3
**Potential Impact:** 3
**Recommendation:** 3
**Confidence:** 4

**Review:**

The coarse-to-fine framework for cross-subscene mobile manipulation presented in this paper is a significant contribution, enhanced by its real-world mobile manipulation experiments.

However, the separation of navigation and manipulation stands out as a critical weakness. Currently, the mobile aspect is just calling off-the-shelf APIs for moving between different tables, while the selling point of the paper is about mobile manipulation. Let's consider we are in a simulation, where perfect sensors (RGBD and camera poses) and a floating gripper are provided, what fundamentally distinguishes this setup from tabletop manipulation setups like F3rm?

I would suggest:
1. Consider navigation as an integral part of the tasks by introducing obstacles between tables. This would require the robot to predict base poses during navigation in addition to end-effector poses, thereby presenting a more comprehensive and realistic problem formulation.
2. Move beyond simplistic, toy-example tabletop setups and explore realistic scenarios where cross-subscene functionality is essential. For inspiration and practical examples, refer to the HomeRobot paper.
3. Also, the paper points out capture failure is the major limitation of the paper. I would suggest the authors try out iPhone cameras, which provide better camera pose tracking and depth than the current setup. Refer to CLIP-Fields, OK-robot, etc. Also, realsense D435 adding T265 is an option.

A lot missing citations in prior mobile manipulation papers need to be considered, including but not limited to:

CLIP-Fields: Weakly Supervised Semantic Fields for Robotic Memory https://arxiv.org/abs/2210.05663

Mobile ALOHA: Learning Bimanual Mobile Manipulation with Low-Cost Whole-Body Teleoperation https://arxiv.org/abs/2401.02117

Adaptive Mobile Manipulation for Articulated Objects In the Open World https://arxiv.org/abs/2401.02117

TeleMoMa: A Modular and Versatile Teleoperation System for Mobile Manipulation https://arxiv.org/abs/2403.07869

**Quality Of The Limitations Section:**

2

**Questions For Rebuttal:**

My major concerns are listed in the reviews above.

Some minor concerns:

“Grounded-Light-HQ-SAM is introduced to extract queries and encode semantics following into Gaussians, which can be queried to provide a coarse pose of the manipulation target. ” More details are needed to explain how you obtain the manipulation target pose.

“Unlike most mobile manipulation methods that only support simple pick-and-place tasks”
This is not true. A lot of prior mobile manipulation papers aer missing in the related work.

The main paper should include more comprehensive details about the task setup, specifically addressing elements such as the definition of task success, as well as the initial configurations of both the robot and the objects involved.

**Robotics Focus:**

4

**Summary Of Paper:**

The paper proposed a Coarse-to-fine framework for cross-subscene mobile manipulation applications.   The paper used a diffusion-based 3D Gaussian completion and inpainting technique to obtain the refined target pose for manipulation.  Real world mobile manipulation experiments are conducted.

**Summary Of Recommendation:**

I expect to see more realistic setup in the real-world experiments.

---

### Author Rebuttal · Authors · 2024-08-14

__The separation of navigation and manipulation__

We thanks for the reviewer's suggestion, but we politely disagree with this point. Actually, one shall notice that this involves the technical roadmap of the whole system. Despite it sounds like an better and idea solution (maybe in the near future), in practice integrating navigation and manipulation together will make our whole system too big to be optimized and inferred. We add the detailed physical setup in the new Supplementary. Our proposed TaMMa leverages cross-subscene fine-grained mobile manipulation in a coarse-to-fine manner, aiming to provide a strong and highly versatile baseline for the mobile manipulation task. Further fine-tuning on specific platforms would be one of our important future directions, but we believe this doesn't diminish our contributions.

__Evaluation on more platforms and comparison with more works__

Indeed, we only use Franka, which is most widely used arm for research. However, our model is built upon the ROS, and we donot have any specific phyical setting that is tightly related to Franka itself. Our model and system shall have very good portability and robust performance across different collaborative robotics. We compare the hardware setups in supplementary Section A and show that we employ standard and widely used hardware. Actually, we are very eager to extend our work to more robotic platforms. However, we currently only have one Franka Panda arm with mobile base that can support mobile manipulation tasks.

During our research, we also discovered that few related works on mobile manipulation are open-sourced or easily reproducible. Additionally, there is a significant gap in the evaluation metrics of these methods. In response to this situation, we have selected representative works: F3RM and Homerobot, and further deployed cross-subscene target-driven mobile manipulation tasks. We are considering releasing the associated codes for TaMMa and sharing some of our deployment experiences.
__More methodology details and additional experiment settings.__

As for the detailed discussion of limitations on challenging objects and environments, we provide additional explanation in supplementary in Section E. We also upload a new video.

As for the detailed diffusion-based depth completion and pose estimation, We provide pseudocode in supplementary Section G. In addition, we hope to release the main part of code TaMMa.

---

### Decision · Program_Chairs · 2024-09-04

**Decision:**

Accept

**Comment:**

The paper introduces a new method for robot mobile manipulation across multiple sub-scenes. The method uses 3D scene reconstruction based on 3D Gaussians, scene inpainting and depth completion to build a framework for mobile manipulation.

Strengths

- The method demonstrates how to leverage perception techniques including 3D Gaussian splatting, depth completion, and scene inpainting for mobile manipulation.
- The framework is evaluated on several real-world mobile manipulation tasks.

Weaknesses

The reviewers have raised several concerns about the paper.

- Reviewer pWYS mentioned the separation of the navigation and the manipulation of the proposed method.
- Missing citations of previous mobile manipulation works.
- Some missing details of the proposed method.

Post-Rebuttal
- The authors have successfully addressed the concerns from the reviewers.